# A High-Certainty Visual Servo Control Method for a Space Manipulator with Flexible Joints

**DOI:** 10.3390/s23156679

**Published:** 2023-07-26

**Authors:** Tao Yang, Fang Xu, Shoujun Zhao, Tongtong Li, Zelin Yang, Yanbo Wang, Yuwang Liu

**Affiliations:** 1State Key Laboratory of Robotics, Shenyang Institute of Automation, Chinese Academy of Sciences, Shenyang 110016, China; xufang@sia.cn (F.X.); liuyuwang@sia.cn (Y.L.); 2Institutes for Robotics and Intelligent Manufacturing, Chinese Academy of Sciences, Shenyang 110169, China; 3University of Chinese Academy of Sciences, Beijing 100049, China; 4Beijing Institute of Precision Mechatronics and Controls, Beijing 100076, China; shoujun.zhao@lasat.com (S.Z.); litongbit@163.com (T.L.); yangzelindream@163.com (Z.Y.); www-2001@163.com (Y.W.); 5Laboratory of Aerospace Servo Actuation and Transmission, Beijing 100076, China

**Keywords:** space manipulator, visual servo, certainty, gradient optimization, adaptive control

## Abstract

This paper introduces a novel high-certainty visual servo algorithm for a space manipulator with flexible joints, which consists of a kinematic motion planner and a Lyapunov dynamics model reference adaptive controller. To enhance kinematic certainty, a three-stage motion planner is proposed in Cartesian space to control the intermediate states and minimize the relative position error between the manipulator and the target. Moreover, a planner in joint space based on the fast gradient descent algorithm is proposed to optimize the joint’s deviation from the centrality. To improve dynamic certainty, an adaptive control algorithm based on Lyapunov stability analysis is used to enhance the system’s anti-disturbance capability. As to the basic PBVS (position-based visual servo methods) algorithm, the proposed method aims to increase the certainty of the intermediate states to avoid collision. A physical experiment is designed to validate the effectiveness of the algorithm. The experiment shows that the visual servo motion state in Cartesian space is basically consistent with the planned three-stage motion state, the average joint deviation index from the centrality is less than 40%, and the motion trajectory consistency exceeds 90% under different inertial load disturbances. Overall, this method reduces the risk of collision by enhancing the certainty of the basic PBVS algorithm.

## 1. Introduction

The visual servo method is crucial for capturing target objects in space, serving as the foundation for future on-orbit facility maintenance [1]. However, the close proximity between the manipulator and the target poses collision risks, potentially damaging delicate components. Therefore, ensuring the safety of the visual servo algorithm is of utmost importance. The main cause of collisions lies in the uncertainty of the relative motion between the manipulator and the target due to external disturbances [2]. Analyzing the visual servo process from a kinematic perspective reveals that the manipulator’s motion trajectory is influenced by the target’s uncertain motion trajectory. Consequently, the manipulator’s motion trajectory can also exhibit uncertainty. For instance, when tracking a target in 3D space, inadequate coordination among the motion trajectories in all directions can give rise to collision risks during the motion process. From a dynamic perspective, the manipulator exhibits flexible characteristics, and there is a possibility of oscillations occurring during high-speed target tracking, which can result in collision risks. Therefore, to enhance the safety of visual servo algorithms, it is necessary not only to design reasonable motion planning strategies to improve kinematic certainty but also to develop robust dynamic controllers. These controllers should be able to mitigate the influence of joint load inertia and stiffness variations on the manipulator during tracking movement, thereby improving the certainty of the dynamics.

### 1.1. Related Works

The position-based visual servo (PBVS) method uses the feedback information of the target position obtained from image processing to guide the motion of the manipulator. It does not suffer from the problem of image Jacobian singularity, leading to higher global stability [3]. Consequently, PBVS algorithms are widely adopted in practical engineering applications. Numerous researchers have put forward their own viewpoints on enhancing the motion control performance of PBVS. Haobin Shi designed a visual servo system with active collision avoidance for a manipulator with redundant joints. The virtual repulsive force function between the manipulator and the target is established, and a safe distance between the manipulator link and the target is always maintained by selecting the appropriate inverse kinematics solution of the redundant manipulator [4]. Tingting Wang proposed a model-based dual-camera visual servo control strategy for circularly symmetric targets with large displacements. According to the distance between the manipulator and the target, the strategy is divided into two stages: the fast-moving stage and the position-tracking control stage; fast and accurate docking can be achieved by adjusting the algorithm models of different stages [5]. The above methods focus on the visual servo motion optimization strategy based on obstacle avoidance and rapidity criteria, but there is still a large uncertainty in the coupling between the motion trajectory and the target trajectory. Some scholars have proposed intelligent algorithm control strategies based on artificial neural networks [6,7,8], synovial membrane variable structure control [9,10,11], and control strategies based on prediction methods [12,13,14]. However, the reliability of these methods may not be suitable for aerospace engineering.

Currently, there is a lack of literature specifically addressing the dynamic control methods for the visual servo process of manipulators. However, there are many studies on suppressing the vibration of flexible space manipulators, which can be used for reference. T. Hao derived the dynamic equation of a general flexible joint manipulator and proposed a static feedback linearization controller to achieve accurate trajectory tracking of the end effector in Cartesian space [15]. M.H. Korayem et al. proposed a design scheme for an SDRE controller and estimator to control a flexible joint manipulator in the presence of noise or external interference [16]. While the method exhibits good universality, it is characterized by high sensitivity to parameter variations. J. Z. Sąsiadek et al. introduced a novel direct adaptive composite controller based on the normalized fuzzy logic system [17]. Wei He proposed a full-state feedback neural network controller [18]. M. Seidi et al. presented a fuzzy-model-based control method for flexible joint robotic arms [19]. Je Sung Yeon et al. proposed a robust controller based on motor-side dynamics [20]. L. Le-Tien addresses a robust adaptive control scheme based on a cascaded structure with a full-state feedback controller with integrator terms as the inner control loop and computed torque as the outer control loop for flexible joint robots [21]. M. A. Ahmad et al. developed a hybrid controller based on a PD-type fuzzy logic control and input shaper. S. E. Talole et al. proposed a trajectory tracking controller for flexible joint robotic systems based on state observer [22]. K. Rsetam et al. proposed a hierarchical sliding mode control (HSMC) design for rotationally flexible joint manipulators (RFJM), proving the closed-loop stability of the system using Lyapunov’s theorem [23,24]. These methods rely less on the accuracy of control model parameters but may involve higher computational complexity. Adaptive control algorithms, with their simple structure and lower computational requirements, offer a promising approach for addressing the dynamic tracking control problem of flexible joint robots [25,26,27]. Considering the short computation cycle in the visual servo process, this paper adopts an adaptive control algorithm to tackle the issue of flexible vibrations in the manipulator.

### 1.2. Motivations and Contributions

Based on the research of previous scholars, this paper proposes a systematic visual servo control strategy for flexible joint space manipulators from an engineering application perspective. The strategy includes position-based visual servo control, motion planning, and dynamic control and aims to address the safety risks of the classical PBVS method. This paper makes contributions to the visual servo control of space manipulators as below.

The method of motion certainty as a key factor in ensuring the safety of visual servo control for space manipulators is introduced in this paper. The safety risks are analyzed from both kinematic and dynamic perspectives, and a high-certainty visual servo control framework is proposed to address these risks.A motion planner is proposed that combines three-stage motion planning in Cartesian space with motion planning in joint space, aiming to minimize deviation from the centrality. The issue of uncertain intermediate states in classical PBVS algorithms is addressed by this method.An adaptive control method for the flexible joint dynamics is proposed in this paper using a Lyapunov-based stability analysis approach. The visual servo control system for space manipulators during dynamic target tracking is, thus, enhanced in robustness.A full-dimensional ground experiment was conducted using a physical mirror method to verify the high-deterministic visual servo control strategy for safety. The experiment was performed in a ground gravity environment, and a simple yet effective method was used to demonstrate the efficacy of the proposed strategy.

The remainder of this paper is organized as follows. In Section 2, the high-certainty PBVS model is established, and the composition of the control system is proposed. The Cartesian space three-stage motion planner, joint space motion optimization method, and dynamic adaptive control algorithm are introduced in detail. In Section 3, a ground experimental system is designed to validate the proposed algorithms in a step-by-step manner using physical mirroring. In Section 4, the robustness of the adaptive control is verified, followed by the safety of the visual servo planner. In Section 5, all the work is summarized, and future research directions are proposed.

## 2. High-Certainty Visual Servo Method

The manipulator’s visual servo system consists of four parts: chaser spacecraft, manipulator, hand-eye camera measurement system, and target spacecraft, as illustrated in Figure 1. The capture process involves two stages. In the first stage, the chaser spacecraft approaches the target and keeps it within a suitable range. In the second stage, the manipulator relies on information measured by the hand-eye camera to capture the target. The visual servo system is a control system with a hand-eye camera as the feedback unit and the manipulator as the execution unit [28,29].

### 2.1. Structure of High-Certainty PBVS

The high-certainty visual servo controller consists of four components: the coordinate converter, the Cartesian-space motion planner, a joint-space motion planner, and the dynamic controller, as depicted in Figure 2.

The coordinate converter maps the target position measured by the camera to the position relative to the manipulator’s base coordinate system based on the manipulator’s configuration. Once the chaser spacecraft approaches the target, the hand-eye camera captures images of the target and then calculates the target’s position and pose matrix, $T_{tar}^{cam},$ relative to the camera coordinate. Then, the target’s position and pose matrix in the TCP (tool central point) coordinate system, $T_{tar}^{tcp},$ is calculated based on the relationship between the camera, the TCP coordinate, and the manipulator. Finally, the target’s position and pose matrix in the manipulator’s base coordinate system, $T_{tar}^{base}$, is computed based on the kinematics of the manipulator. The conversion method is presented in Equation (1).
(1)$$T_{tar}^{base}=T_{tcp}^{base}\times {T_{cam}^{tcp}\times T}_{tar}^{cam}$$

The Cartesian space motion planner controls the motion state of the visual servo process and keeps the space manipulator in a certain state to the target, thereby mitigating the risk of uncertainty. See Section 2.2 for details. The joint space motion planner takes advantage of the redundant degrees of freedom of the manipulator to optimize the allocation of joint positions, avoiding movement to limit positions and ensuring a large safety margin for the manipulator. See Section 2.3 for details. The dynamic controller, also known as the joint servo driver, calculates the expected driving torque. The dynamics controller optimizes the control strategy to handle the effects of external disturbances during the tracking process and maintains the robustness of the manipulator’s dynamic model. See Section 2.4 for details.

### 2.2. Cartesian Space Planner

The visual servo algorithm operates in the manipulator’s base coordinate system. A three-stage motion planning strategy in Cartesian space, building upon the classical PBVS method, is introduced to gradually reduce the distance between the manipulator and the target, as shown in Figure 3.

At the beginning of the visual servo process, the direction with the largest error is selected as the feeding direction, while the other two directions are treated as non-feeding directions. Figure 3 shows the Z direction as the feeding direction. However, motion coupling between different directions can generate uncontrollable trajectories that may result in collisions. To address this issue, the Cartesian space motion planner aims to reduce the motion coupling. The basic idea is to plan an intermediate point that controls the error between the manipulator’s end effector and the target in the non-feed direction to an acceptable range before executing the capture action in the feed direction.

The first stage is the aiming and alignment stage. In this stage, the manipulator tracks steadily in the non-feeding direction while maintaining the current position in the feeding direction. The allowable error thresholds in all three directions are set to ${∆}_{1}$, and the continuous holding time is greater than five measurement cycles of the hand-eye camera. Once these conditions are met, the visual servo process transitions to the second stage.

The second stage is the feed tracking stage. In this stage, the manipulator maintains stable tracking in the X-Y-Z directions. The expected target position in the feeding direction is denoted as ${fed}_{exp2}$, while the expected target position in the non-feeding direction is 0. The allowable errors in all three directions are set to ${∆}_{2}$, and the holding time is required to be longer than 10 hand-eye camera measurement cycles. Once these conditions are met, the system transitions to the third stage for quick capture.

The third stage is the fast capture stage. In this stage, the manipulator performs stable tracking in the three directions of X-Y-Z. The expected position in the feed direction is denoted as ${fed}_{exp3}$, with an allowable error of ${∆}_{fed3}$. The expected value in the non-feed direction is 0, with an allowable error of ${∆}_{ufed3}$. When these conditions are satisfied, the end effector is immediately closed to execute the capture operation.

To ensure reliable capture, the motion planning parameters must satisfy the following relationships:(2)$${∆}_{1}\;\geq \;2\;*\;{∆}_{2}>\;5\;*\;{∆}_{fed2}$$

### 2.3. Joint Space Planner

The manipulator with redundant degrees of freedom has infinite kinematic inverse solutions, theoretically. Numerical iterative methods can be employed to calculate the inverse kinematics, working as joint-space motion planners. A typical calculation approach using the pseudoinverse of the Jacobian matrix is used to determine the joint velocities corresponding to the desired TCP pose velocities. Integrating these joint velocities yields the next cycle’s joint positions [30]. Defining $x_{d}(s)$ as the desired TCP trajectory, $\dot{s}(t)$ as the velocity along the desired trajectory, q as the current joint position vector, J as the Jacobian matrix, and $J^{†}$ as the pseudoinverse of J, the calculation process is shown as Equations (3) and (4).
(3)$$\dot{q}(t_{k+1})=J^{†}\left(\dot{x_{d}}\right)+(I-J^{†}J)\dot{q}(t_{k})$$
(4)$$q\left(t_{k+1}\right)=q\left(t_{k}\right)+\dot{q}(t)∆t$$

The above iterative process can control the joint motion trajectory so that the joint position–related objective function *H* is optimal, as shown in Equation (5).
(5)$$\mathrm{Min}\;H(q\left(t_{i}\right))$$

To ensure faster convergence [31], the iterative process is expected to follow the negative gradient direction ▽H of the objective function H. This transforms the problem of solving the joint angular velocity $\dot{q}(t_{k})$ of the manipulator into a matrix least two-norm optimization problem, allowing the joint velocity and the gradient vector of the *H* function to be as close as possible.
(6)$$\mathrm{Min}\;\left\|\dot{q}\left(t_{k}\right)-k*▽H(q(t_{i+1}))\right\|2$$
(7)$$q\left(t_{k+1}\right)=q\left(t_{k}\right)+\dot{q}(t)∆t$$
(8)$$\dot{q}\left(t_{k+1}\right)=M*\dot{s}(t)+N*\dot{q}(t_{k})$$
(9)$$\dot{x_{d}}=\frac{d_{x}}{d_{s}}*\dot{s}(t)$$
(10)$$M=J^{†}\frac{d_{x}}{d_{s}}$$
(11)$$N=(I-J^{†}J)$$

The minimum value of the least two-norm equation above satisfies the following constraints:(12)$$\left(\begin{array}{c} \dot{s_{k}}\\ u_{k} \end{array}\right)=-\frac{1}{2}k{\left[\begin{array}{cc} {{M_{k}}^{T}M}_{k} & {{M_{k}}^{T}B}_{k}\\ {{B_{k}}^{T}M}_{k} & {{B_{k}}^{T}B}_{k} \end{array}\right]}^{-1}\left[\begin{array}{c} {M_{k}}^{T}∆H(q)\\ {B_{k}}^{T}∆H(q) \end{array}\right]$$
(13)$$N=[0,B_{k}]Q_{k}$$
(14)$$\left[\begin{array}{c} u_{k}\\ w_{k} \end{array}\right]=Q_{k}\dot{q}(k)$$

Among them, $Q_{k}$ is a column variable square matrix with LU decomposition. The following equation is the iterative algorithm equation of the gradient descent method of redundant manipulators.
(15)$$\dot{q}\left(t_{k+1}\right)=M*\dot{s}(K)+B*U_{K}$$

The joint space motion planning steps can be summarized as follows.

(1)Calculate the Cartesian velocity from the starting position and the target position $\frac{d_{x}}{d_{s}}$ and the number of iterative steps.(2)Calculate the Jacobi matrix for the current joint angle *q* configuration J and the kinematics result *x(q)*.(3)According to the equation **M =**
$J^{†}\frac{d_{x}}{d_{s}}$, **N =**
$\left(I-J^{†}J\right),$ calculate **M_k_** and **N_k_**.(4)Do LU decomposition of **N** matrix and calculate **B_k_** and $Q_{k}$.(5)Calculate $\dot{s}\left(k\right)$ and **U_k_** from **M_k_, B_k_,**
$Q_{k}$.(6)Calculation $\dot{q}\left(t_{k+1}\right)=M*\dot{s}(k)+B*U_{K}$.(7)Repeat steps 2 to 7 until the iterative steps are completed.

### 2.4. Lyapunov-Based Adaptive Joint Dynamic Controller

#### 2.4.1. Dynamic Adaptive Controller Structure

The space manipulator has a large structural span, and the joint’s transmission is generally designed with light and flexible materials. As shown in Figure 4, during the motion, it exhibits significant flexibility characteristics. Therefore, the space manipulator joint can be considered a nonlinear dynamic system with flexible joints. When the manipulator tracks the target using visual guidance information, the arm shape undergoes significant changes, and the load inertia and transmission stiffness of each joint also vary greatly. This changes the parameters of the joint control system, leading to model perturbations and causing the joint’s motion response to deviate from the expected state. In severe cases, this can result in chatter [32].

The engineering feasibility is taken into account by simplifying the flexible joint of the space manipulator as a spring-damper system with stiffness K and viscous friction f, as shown in Figure 5. The simplified functional model of the joint position input $\theta_{i}\left(s\right)$ and position output $\theta_{o}\left(s\right)$ is given below:(16)$$\frac{\theta_{o}\left(s\right)}{\theta_{i}\left(s\right)}=\frac{k}{Js^{2}+fs+k}$$

The undamped oscillation frequency $\omega_{n}$ and the system damping ξ of the joint dynamics model are calculated as follows:(17)$$\omega_{n}=\sqrt{\frac{k}{J}}$$
(18)$$\xi =\frac{f}{2\sqrt{Jk}}$$

According to the above equation, an increase in joint load inertia or a decrease in joint stiffness will lead to a decrease in system damping and a decrease in the undamped natural oscillation frequency. Since the fundamental frequency of the space manipulator structure is generally low, this can easily lead to overshoot or even instability of the system. In order to achieve effective joint control, it is necessary to control the stiffness and damping of the joint dynamics control system within the expected range. Therefore, the joint dynamics model reference adaptive controller shown in Figure 5 has been designed.

#### 2.4.2. Space Manipulator Dynamics Model

The 7-DOF space manipulator dynamics model is as follows:(19)$$\tau =D\left(q\right)\ddot{q}+H(q,\dot{q})$$

The joint position and velocity are taken as state vectors, and the dynamics model is transformed into a state space model.
$$x={\left[q,\dot{q}\right]}^{T}$$
(20)$$\dot{x}=Ax+B\tau$$
where the coefficients of the state space equation are:(21)$$A=\left[\begin{array}{cc} O & I\\ 0 & -D^{-1}H \end{array}\right],\;B=\left[\begin{array}{c} 0\\ D^{-1} \end{array}\right]$$

#### 2.4.3. Reference Joint Dynamics

Assuming a constant system of joint flexibility link stiffness, the kinetic equation for the joint position is established as:(22)$$J\ddot{\theta_{o}}+f\dot{\theta_{o}}+k\theta_{o}=k\theta_{i}$$

The joint position and velocity are taken as state 2 × 2 dimensional vectors, and the differential equations of joint dynamics are transformed into a state space model.
(23)$$y={\left[\theta_{o},\dot{\theta_{o}}\right]}^{T}\;\dot{y}=A_{m}y+B_{m}\theta_{i}$$
where the state space equation coefficients are:(24)$$A_{m}=\left[\begin{array}{cc} O & I\\ -\frac{k}{J} & -\frac{f}{J} \end{array}\right]B_{m}=\left[\begin{array}{c} 0\\ \frac{k}{J} \end{array}\right]$$

#### 2.4.4. Control Lawyer

Establish the control input equation, which consists of two parts: state feedback and motor angle feedforward.
(25)$$\tau =-k_{x}x+k_{r}\theta_{i}$$
where $k_{x}$ is the state feedback coefficient, and $k_{r}$ is the input feedforward coefficient.

Substitute the input equation into the dynamic state equation of the manipulator
(26)$$\dot{x}=Ax+B\left(-k_{x}x+k_{r}\theta_{i}\right)=(A-Bk_{x})x+Bk_{r}\theta_{i}=A_{s}x+B_{s}\theta_{i}$$

Among them
(27)$$A_{s}=\left[\begin{array}{cc} O & I\\ -D^{-1}k_{x1} & -D^{-1}(H+k_{x2}) \end{array}\right]$$
(28)$$B_{S}=\left[\begin{array}{c} 0\\ D^{-1}k_{r} \end{array}\right]$$

The above equation is the closed-loop control state model of the space manipulator’s arm joint. Define the system’s systematic error as shown in the following equation.
(29)e=y−x
(30)$$\dot{e}=A_{m}e+\left(A_{m}-A_{s}\right)x+\left(B_{m}-B_{s}\right)\theta_{i}$$
(31)$$\dot{e}=\left[\begin{array}{cc} O & I\\ -\frac{k}{J} & -\frac{f}{J} \end{array}\right]\;e+\left[\begin{array}{cc} O & 0\\ D^{-1}k_{x1}-\frac{k}{J} & D^{-1}\left(H+k_{x2}\right)-\frac{f}{J} \end{array}\right]x+\left[\begin{array}{c} 0\\ \frac{k}{J}-D^{-1}k_{r} \end{array}\right]\theta_{i}$$

#### 2.4.5. Adaptive Layer

Calculation of the purpose of adaptive control $k_{x}\;and$ $k_{r}$ parameter control rate such that the error e converges to 0, i.e.,
(32)$$\underset{t\to \infty }{\lim}e(t)=0$$
constructing the following Lyapunov function from the joint closed-loop control state equation [33].
(33)$$V=e^{T}Pe+tr\left({\left(A_{m}-A_{s}\right)}^{T}F_{A}^{-1}\left(A_{m}-A_{s}\right)\right)+tr\left({\left(B_{m}-B_{s}\right)}^{T}F_{B}^{-1}\left(B_{m}-B_{s}\right)\right)$$
(34)$$\dot{V}=e^{T}\left({A_{m}}^{T}p+pA_{m}\right)e+tr\left({\left(A_{m}-A_{s}\right)}^{T}\left(pex-F_{A}^{-1}\dot{A_{s}}\right)\right)\;+tr\left({\left(B_{m}-{AB}_{s}\right)}^{T}\left(pe\theta_{i}-F_{B}^{-1}\dot{B_{s}}\right)\right)$$

According to Lyapunov stability theory, V is a negative definite matrix, then
(35)$${A_{m}}^{T}p+pA_{m}=-Q=-I$$
(36)$$\dot{A_{s}=}F_{A}pex\approx B\dot{k_{x}}$$
(37)$$\dot{B_{s}=}F_{B}pe\theta_{i}\approx B\dot{k_{r}}$$

In turn, the adaptive control rate is:(38)$$\dot{k_{x}}={k_{x}{B_{m}}^{-1}F}_{A}pex$$
(39)$$\dot{k_{r}}={k_{r}{B_{m}}^{-1}F}_{B}pe\theta_{i}$$

## 3. Experiment Design

The space manipulator operates in a microgravity environment with small joint torques, making it unable to work in a gravity environment on the ground. To verify the effectiveness of the visual servo control strategy for the space manipulator in orbit, a physical mirror experiment system was designed. The purpose of this mirror system is to map the joint motion state of the space manipulator, which cannot move “upright” onto the mirror manipulator in real-time, allowing for the reproduction of the space manipulator’s three-dimensional motion in a gravity environment [34].

Figure 6 shows the composition of the physical mirror experiment system, which consists of a space manipulator mounted on air bearing equipment, a mirror manipulator fixed on the ground, a hand-eye camera, and a target simulator. The target simulator is driven by an industrial manipulator, simulating the movement of the target in orbit. The mirror manipulator has the same kinematic parameters as the space manipulator, but the joint drive torque is large enough to overcome gravity and enable movement on the ground. The mirror experiment system operates on the following principles.
Joint motion decoupling: The space manipulator is mounted on a marble air bearing platform. First, the manipulator is decoupled into two independent parts by fixing the “elbow” structure, which consists of a 3-degree-of-freedom mechanism and a 4-degree-of-freedom mechanism. Air bearing devices are then designed for the two manipulators with fewer degrees of freedom, allowing all joints to rotate. Figure 7 illustrates the principle of this process.


Joint motion mirroring: The motion state, including position and speed information, of the joints of the space manipulator is transmitted in real-time through the fieldbus network to the mirror manipulator. The mirror manipulator drives its joints to track the received motion state in real time, reproducing the three-dimensional motion of the space manipulator.Closed-loop information link: The mirror manipulator moves with the hand-eye camera (which is mounted on the mirror manipulator). The hand-eye camera captures the target position and feeds it back to the space manipulator visual servo controller via the fieldbus. The controller then uses the visual servo algorithm to calculate the error, complete the motion planning and dynamics calculation, and generate joint motion instructions to drive the joints of the space manipulator on the air bearing platform. The system information link is shown in Figure 8.


### Experimental Parameter Settings

The CAN2.0B standard [35] is applied to the network of the physical mirroring experimental system (Figure 9), with the manipulator serving as the master node and the space manipulator, the mirror manipulator, and the hand-eye camera serving as slave nodes. The range of motion for each joint is between −170° and 170°. Table 1 shows the control parameters of the three-stage motion planning strategy used in the experiment.

The objective optimization function used in the experiment for joint space is the joint deviation center index, and its calculation method is shown in the following equations:(40)$$H\left(i\right)=1-\frac{\left(q_{i}^{max}-q_{i}\right)\left(q_{i}-q_{i}^{min}\right)}{{\left(q_{i}^{max}-q_{i}^{min}\right)}^{2}/4},\;i=1:7\;$$

The visual servo algorithm verification experiment is conducted in two stages. The first stage verifies the effectiveness of the dynamic adaptive control algorithm (details can be found in Section 4.1), while the second stage verifies the correctness of the entire high-certainty visual servo algorithm (details can be found in Section 4.2).

## 4. Discussion

### 4.1. Dynamic Adaptive Experiment

The dynamic adaptive control experiment is conducted by comparing the consistency of joint motion response when the manipulator captures the same target under different disturbance conditions. The specific steps of the experiment are as follows:

Step 1: Install the space manipulator on the air bearing device and keep joints 2, 4, and 6 in a microgravity state. This test method is called the planar motion method, as shown in Figure 10. Before conducting the experiment, the height of the captured target is adjusted to be the same as that of the manipulator to enable the coordinated movement of the above three joints. This allows the manipulator to complete the visual servo action on the marble plane and record the motion response data of the three joints.

Step 2: Carry out the capture experiment with the mirror experiment system as shown in Figure 8 (the load condition of joints 2, 4, and 6 is very different from that in Figure 11), and record the motion response data of the three joints.

Step 3: Compare the motion data of the three joints in Steps 1 and 2 as the quantitative analysis of the effectiveness of the dynamic adaptive algorithm. If the motion curves in Steps 1 and 2 are substantially similar, it indicates that the joint adaptive control algorithm possesses robust anti-disturbance capabilities. The effectiveness of the dynamic adaptive control algorithm was quantified by calculating the difference in the sum of the absolute values of joint dynamic tracking errors. The evaluation criteria are defined as below:(41)$$q_{err}=\int_{t0}^{t1}\left|q_{cmd}-q_{fed}\right|dt$$

The left parts of Figure 11, Figure 12 and Figure 13 are position response curves and tracking error curves of joints 2, 4, and 6 for the plane air bearing experiment, and the right parts are the joints of the physical mirroring experiment. The curve marked as command is the joint position command, the curve marked as feedback is the actual joint position, and the curve marked as following error is the difference between the joint position, command position, and actual position. The curves show that the joint response curves of the two methods are relatively consistent.

Table 2 counts the cumulative tracking errors of the three joints under the two methods according to Equation (41) and calculates the closeness of the physical mirroring method based on the air bearing method. The experimental data show that the cumulative tracking error of joint 2 is 40.38° when the air bearing method is used, and the cumulative tracking error of joint 2 is 39.3° when the physical-mirroring method is used. For joint 2, the cumulative tracking error of the physical-mirroring method is 98.6% based on the cumulative tracking error of the air bearing method as a reference (100%), indicating that the tracking characteristics of the two methods are very consistent, with a consistency of 98.6%. Similarly, for joint 4 and joint 6, the consistency of the two methods is 99.7% and 93.2%, respectively. The experimental results show that although the load characteristics of joint 2, joint 4, and joint 6 are very different when using the air bearing method, there is basically no difference in the motion response characteristics. These data prove that the robustness of the dynamic model reference adaptation algorithm is very good. With the support of this algorithm, the influence of the change of inertia on the motion characteristics of the manipulator can be ignored. Therefore, the physical-mirroring method can be used in the Cartesian space target capture experiment.

### 4.2. Whole Process Verification of the Algorithm

The whole process verification experiment of the visual servo algorithm was performed using the physical mirror experiment system to execute the high-deterministic visual servo algorithm. The target simulator controlled the captured target to move randomly in the directions X or Y or Z at a speed of less than 50 mm/s. The manipulator approached and captured space targets under the guidance of the hand-eye camera. During the experiment, the measured values of the target position of the hand-eye camera and the motion responses of the 7 joints of the manipulator were recorded. The experimental process is depicted in Figure 14.

Figure 15, Figure 16 and Figure 17 depict the entire visual servo process, showing the position of the target relative to the manipulator’s TCP coordinate system as measured by the hand-eye camera. The position vector of the target relative to the TCP coordinate system of the manipulator at the start of the experiment is [78.4, 90.7, 590.7] mm, which satisfies the error control requirements of the first stage of the three-stage planning strategy. As the manipulator tracks the target in three directions, the relative error decreases, and at 14.85 s, the error is less than 50mm, allowing the system to enter the second stage. The position vector of the target relative to the TCP coordinate system of the manipulator at the beginning of the second stage is [66.8, 76.9, 590.7] mm. In the second stage, the manipulator continues to track the target and reduces the relative error until 21.1 s, when it satisfies the error control requirements of the third stage. At this point, the capture operation is executed, and the visual servo process ends. The position vector of the target relative to the TCP coordinate system of the manipulator at this time is [−6.5, 23.7, 154.2] mm. Table 3 shows the relative position data at each stage, demonstrating that the manipulator follows the three-stage active motion planning strategy and exhibits strong Cartesian space motion certainty.

According to the visual servo strategy, the manipulator completes the sequential switching of the three stages with no significant fluctuation. Figure 18 shows the motion response curves of the joints of the manipulator during the entire experiment. The curve labeled “cmd” represents the position command of the joints, and the curve labeled “fed” represents the actual position feedback of the joints. Since the visual servo strategy directly adopts a formal joint position command, the error between the command and feedback is relatively large. However, due to the application of the dynamic adaptive algorithm, the movement of the manipulator remains stable even under the action of a large step command, and no obvious vibration is observed. The statistics of the deviation center index of each joint are presented in Table 4. The data indicate that the average deviation center index of the seven joints is 0.3986, indicating that the average movement margin of the joints of the manipulator is relatively large. Therefore, the joint space motion planning algorithm has achieved the expected objective.

## 5. Conclusions

This paper proposes a high-certainty visual servo control framework for a space manipulator. The framework is designed to address the uncertainty of the motion, which can lead to collision during the process. The framework includes a three-stage motion planning strategy in Cartesian space and motion optimization in joint space using the gradient descent method. Additionally, an adaptive control algorithm for space manipulator dynamics based on Lyapunov stability analysis is proposed to reduce the interference of model parameter changes on the motion control performance and improve the robustness of the manipulator. The experimental results show that the high-certainty algorithm has a motion response consistency of 93.2% in the presence of external inertia disturbance, demonstrating strong robustness to the perturbation of the model. The visual servo experiment successfully completed the three-stage switching of the Cartesian space, effectively improving the certainty of the intermediate state of the visual servo. The average value of the joint deviation center index was 0.3986, indicating that the joints have a strong safety margin.

The results of comprehensive experiments indicate that the proposed high-certainty visual servo control strategy for flexible space manipulators with redundant joints is effective in enhancing the certainty of the visual servo process. Although this paper analyzes the impact of the visual servo algorithm on security from the perspective of kinematics and dynamics and proposes corresponding solutions, the coupling analysis of dynamics and kinematics control algorithms has not yet been carried out. Therefore, future studies will focus on investigating their coupling relationship and further improving the security of the visual servo method.

## Figures and Tables

**Figure 1 sensors-23-06679-f001:**
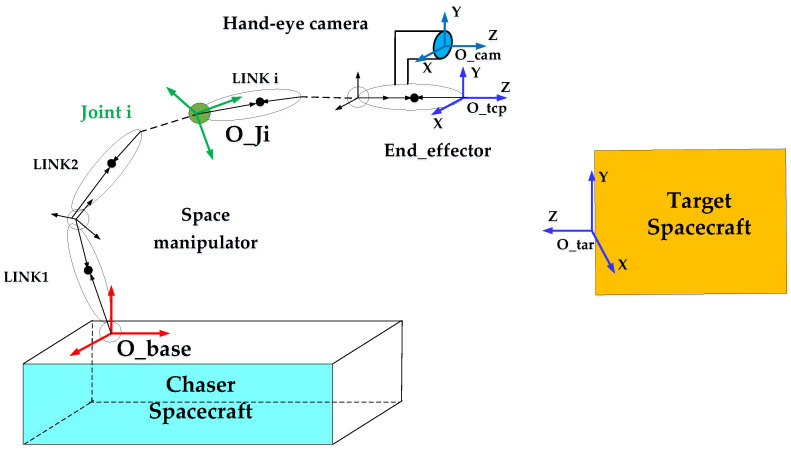
Schematic diagram of the visual servo system of the space manipulator.

**Figure 2 sensors-23-06679-f002:**
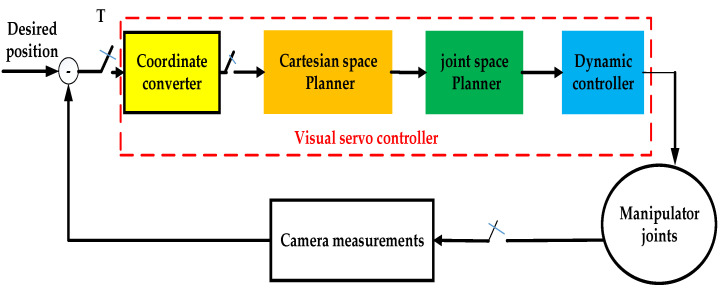
Block diagram of high-certainty PBVS control principle.

**Figure 3 sensors-23-06679-f003:**
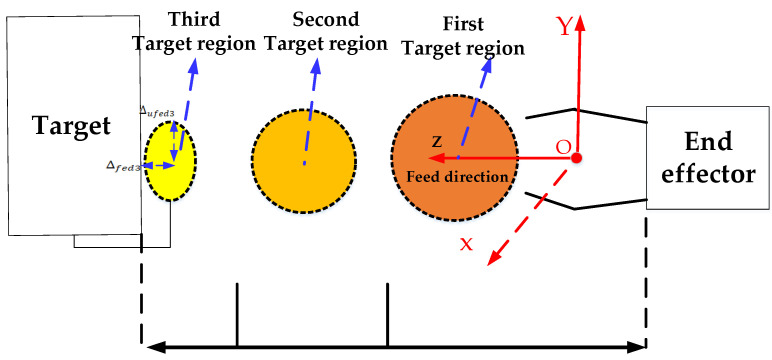
Schematic diagram of three-stage motion planner in Cartesian space.

**Figure 4 sensors-23-06679-f004:**
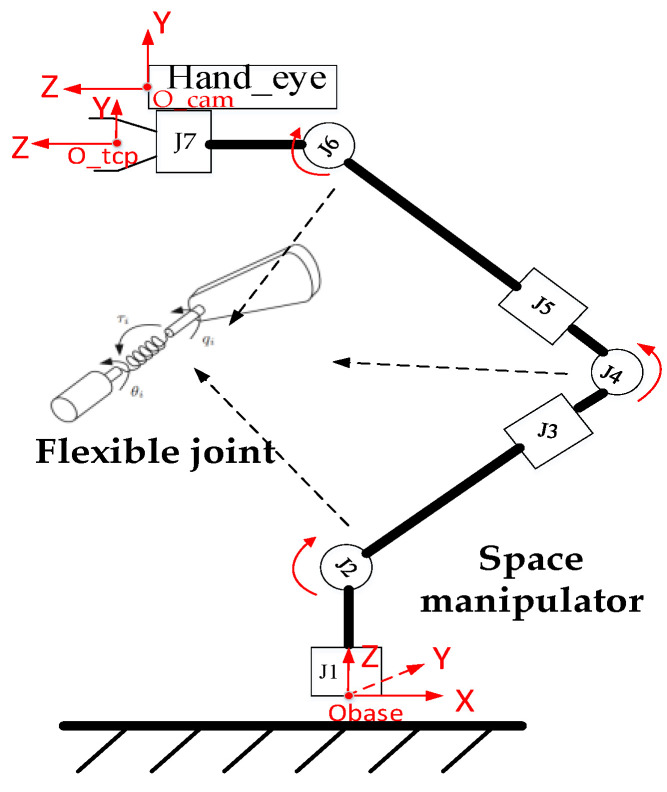
Schematic diagram of a space manipulator with flexible joints.

**Figure 5 sensors-23-06679-f005:**
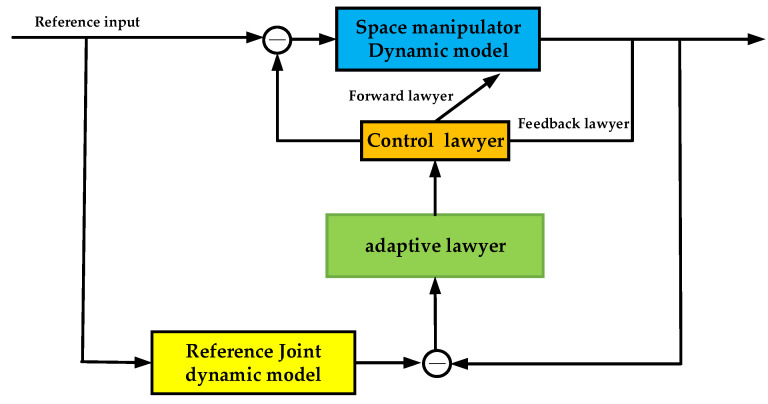
The joint dynamics model reference adaptive control block diagram.

**Figure 6 sensors-23-06679-f006:**
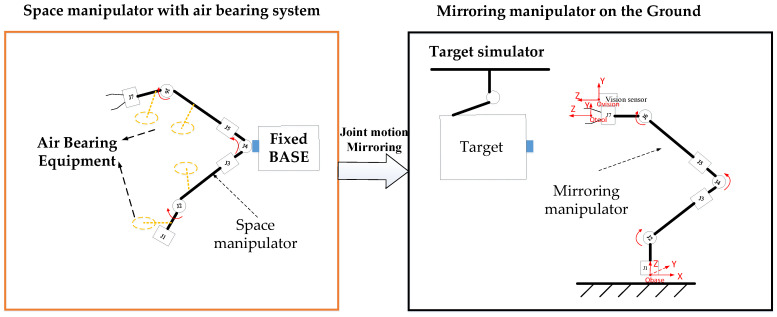
Composition of the physical mirroring system.

**Figure 7 sensors-23-06679-f007:**
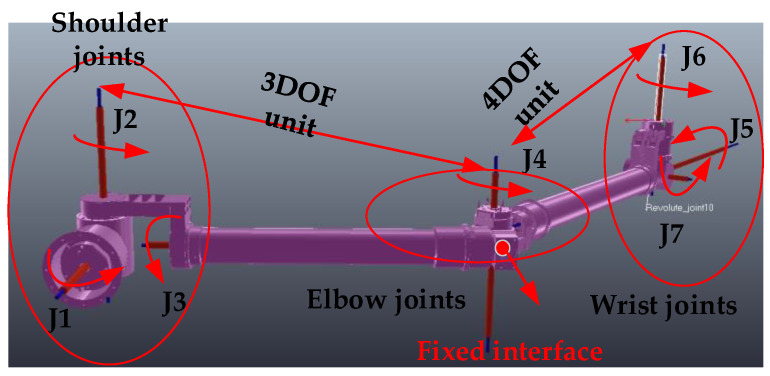
Joint decoupling device with air bearing.

**Figure 8 sensors-23-06679-f008:**
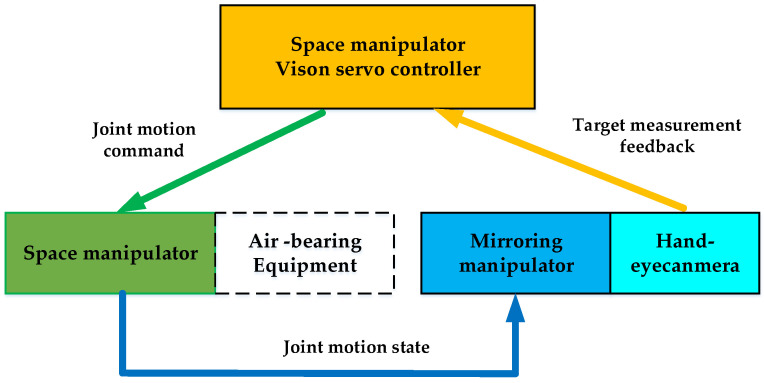
Information flow diagram of physical mirroring experiment system.

**Figure 9 sensors-23-06679-f009:**
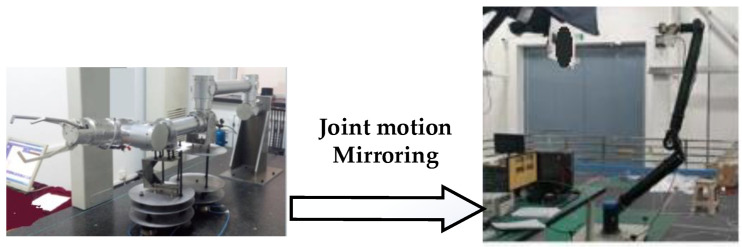
Physical mirror experiment system.

**Figure 10 sensors-23-06679-f010:**
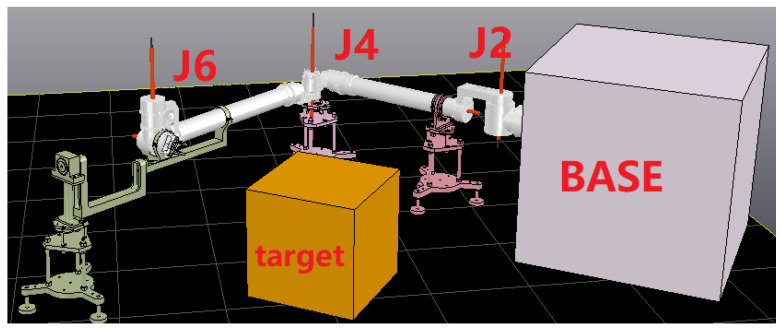
Air bearing experiment on plane.

**Figure 11 sensors-23-06679-f011:**
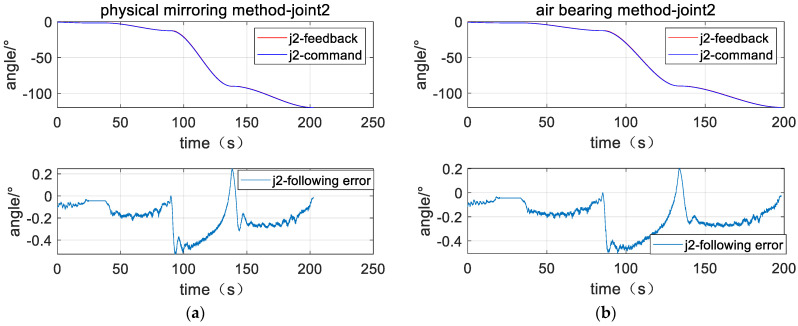
Position response curve of Joint 2 (**a**) plane air bearing experiment; (**b**) physical mirroring experiment.

**Figure 12 sensors-23-06679-f012:**
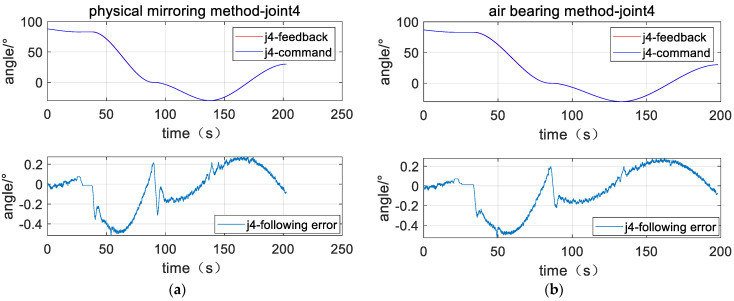
Position response curve of Joint 4 (**a**) plane air bearing experiment; (**b**) physical mirroring experiment.

**Figure 13 sensors-23-06679-f013:**
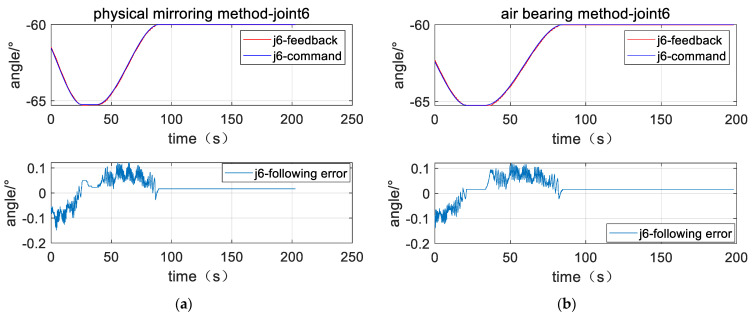
Position response curve of Joint 6 (**a**) plane air bearing experiment; (**b**) physical mirroring experiment.

**Figure 14 sensors-23-06679-f014:**
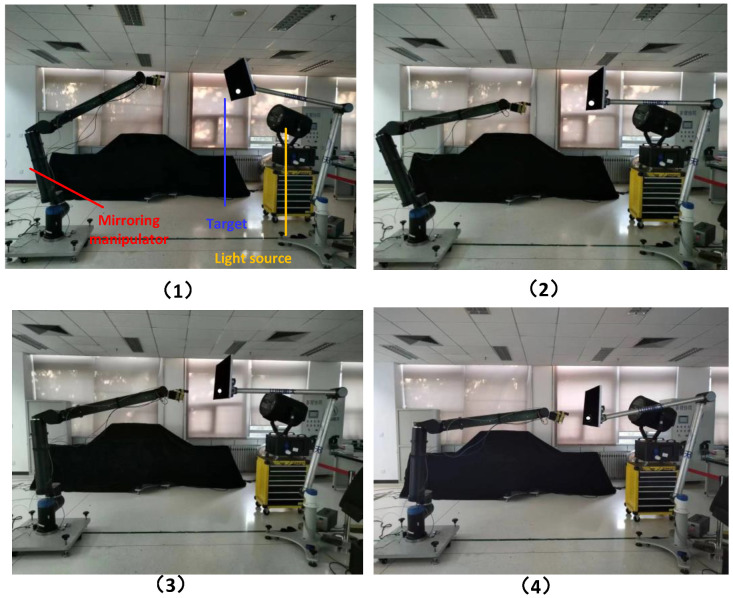
Typical process of the visual servo experimental. (**1**) the beginning; (**2**) (**3**) the intermediate process; (**4**) the end state.

**Figure 15 sensors-23-06679-f015:**
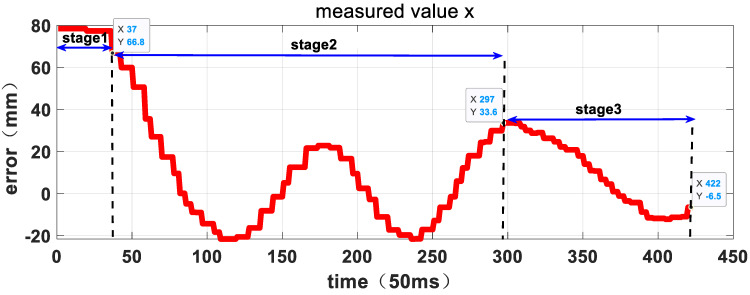
The relative position curve in X direction measured by hand-eye camera.

**Figure 16 sensors-23-06679-f016:**
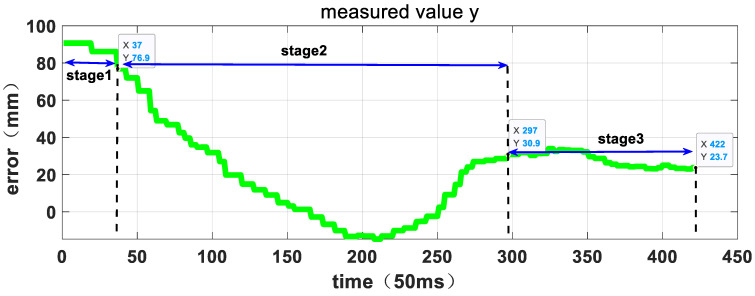
The relative position curve in Y direction measured by hand-eye camera.

**Figure 17 sensors-23-06679-f017:**
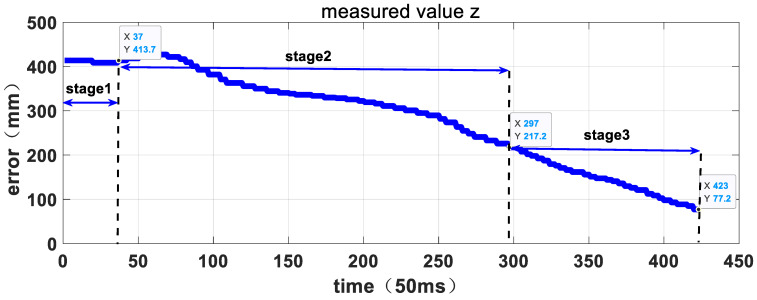
The relative position curve in Z direction measured by hand-eye camera.

**Figure 18 sensors-23-06679-f018:**
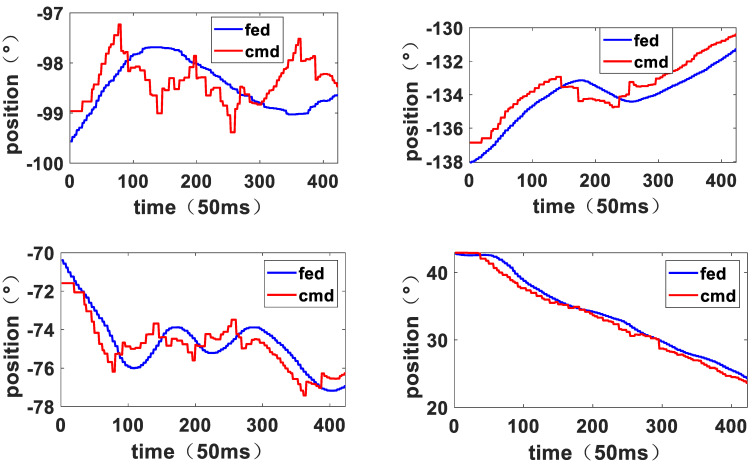
Response curve of joint 1 to joint 7.

**Table 1 sensors-23-06679-t001:** Parameters of the motion planning strategy.

${∆}_{1}$	${∆}_{2}$	${∆}_{fed3}$
100 mm	50 mm	10 mm
${fed}_{exp2}$	${fed}_{exp3}$	${∆}_{ufed3}$
370 mm	240 mm	50 mm

**Table 2 sensors-23-06679-t002:** Total following error.

Method	Joint 2	Joint 4	Joint 6
Physical-mirroring	39.83°	33.84°	6.45°
Air bearing	40.38°	33.93°	6.92°
Percent	98.6%	99.7%	93.2%

**Table 3 sensors-23-06679-t003:** Statistical results of the actual position with the algorithm.

States	Deviation x (mm)	Deviation y (mm)	Deviation z (mm)
Expectations	0	−30	80
Initial value	−78.4	−90.7	413.7
Stage1 to Stage2	−66.8	−76.9	413.7
Stage2 to Stage3	33.6	−30.9	217.2
The finished state	−6.5	−23.7	77.2

**Table 4 sensors-23-06679-t004:** The index of joints deviation center.

Joint 1	Joint 2	Joint 3	Joint 4
0.3357	0.6213	0.1946	0.0311
Joint5	Joint6	Joint7	Average
0.3530	0.4340	0.8206	0.3986

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
