# Peer review of "A High-Certainty Visual Servo Control Method for a Space Manipulator with Flexible Joints"

_sensors, 2023, doi:10.3390/s23156679_

Round 1

Reviewer 1 Report

This paper introduces a novel high-certainty vision servo algorithm for space manipulator with flexible joints, which incorporates an active motion planner and a Lyapunov dynamics model reference adaptive controller. Moreover, a planner in joint space based on the fast gradient descent algorithm optimizes the joint deviation. To improve dynamic certainty, an adaptive control algorithm based on Lyapunov stability analysis, is used to enhance the system's anti-disturbance capability. The paper is well-structured and provides significant contributions to the field of control method for redundant space manipulator. However, there are some areas that require improvement, particularly in the clarity of technical details and the presence of grammar and spelling issues.

(1) The manuscript contains some grammar and spelling errors that detract from its overall readability. A thorough proofreading and editing process is necessary to correct these issues and enhance the manuscript's clarity and presentation. 

(2) The mathematical symbols described in the text should be consistent with those in the figure. For example, in figure 4 “J+”.

(3) Fig. 4 is a schematic diagram of the operation process of the joint space planner, but the mathematical symbols in the diagram are not explained and there is no arrow symbol indicating the data flow direction, which makes it difficult for readers to understand its content.

(4) The mathematical symbols that appear for the first time in the text formulas should be described accordingly.

(5) The ordinal number format in the joint space motion planning steps should be consistent and 7) is a description of the end of the loop not a judgment condition of the end of the loop that can't be put into the loop.

  To sum up, there are some minor errors in the article. The authors should correct the whole article carefully.

The manuscript contains some English grammar and spelling errors that detract from its overall readability. A thorough proofreading and editing process is necessary to correct these issues and enhance the manuscript's clarity and presentation.  

Author Response

Thanks to dear expert for your meticulous guidance. Your guidance is very specific. I accept and express my gratitude for all the revisions you marked, and they have been revised one by one.

Reviewer 2 Report

The authors have done in-depth study and experimental work. The results are satisfactory. There are few corrections that the author can implement.

1. Abstract can be improved as the need for the proposed methodology must be explained clearly.

2. English needs to be improved. Kindly check the entire manuscript

3. The kinematic diagram Fig 1 can be improved by mentioning x y z axis clearly along with the transformation

4. Check the figure captions of the MDPI format and change it accordingly

5. Fig 3 can be improved as the letters inside the figure cannot be read

6. In some places equations are bold maintain consistency and equations will not be in bold.

7. 2.3. Joint Space Planner check the font (line 222)

8. Fig 4 needs improvement

9. Fig 5 is not clear

10. Mention the details how the author created Fig 8

11. Fig 9 needs improvement. Please check the font and font size for all figures throughout the manuscript

12. 3.2 Experimental Parameter Settings (line 398) check font

13. It is better to put the DH parameters table 1 while explaining the kinematic diagram

14. Check the format of the table as the font sizes and fonts are varying. The presentation of the manuscript needs to be improved

15. What is fed and cmd in figure 19.

16. The authors can compare their proposed methodology with the existing work.

Needs moderate improvement.

Author Response

(The authors gave the same response as above.)

Reviewer 3 Report

I congratulate authors for performing this interesting control engineering based study for the manipulators with flexible joints. However, there are following suggestions which authors need to go through: 

1. Fig 10b is not very clear. 
2. Fig 15 is not very clear and do numbering of the images and define all of them in the caption/text.
3. Authors need to fully revise their introduction section. The introduction section seem huge and not very concise.
4. Section 2 of the manuscript (i.e. high-certainty servo vision method) also seem quite large. Reduce and concise the same, appropriately.

5. I would like to know more about the mathematical explanation of section 2.3 and 2.4. Is this explanation performed by author itself or they have taken some assistance/help from any reference? 

Author Response

(The authors gave the same response as above.)

Reviewer 4 Report

The article deals with the interesting topic of redundant space manipulator control with flexible joints.

The name is consistent with the content, but the article does not provide more detailed information about the properties of the manipulator (e.g. weight, material used, dimensions, camera parameters, etc.).

The used literature is sufficient, its analysis is primarily devoted to the Introduction chapter.

Chapter 2 is processed at a good level except for a few formal errors and the quality of the fig. 3 and 5. The descriptions in the pictures are not clearly visible.

The third chapter is focused on Experiments design. The chapter is processed at a good level up to fig. 7 and 10.

The fourth chapter presents measured data that confirm the correctness of the chosen manipulator control method.

The post contains typographical and grammatical inaccuracies and a few spelling errors, but these issues should be resolved with the help of a good editor.

I mainly ask the authors to check and correct the following parts of the article:

1. Line 344, please replace the character () with the equivalent in English.

2. Correct the error in the description of Figure 7.

3. It is necessary to unify the labelling of variables in formulas (eg italics).

4. It would perhaps be more appropriate to bring out the descriptions of the components from fig. 10 on the desktop near the picture (for clarity).

5. The meaning of the sentences in lines 453 and 454 is not clear to us.

6. The title of chapter 4.2 (line 466) should begin with a capital letter.

7. Image 15, quality is insufficient.

Author Response

(The authors gave the same response as above.)

Round 2

Reviewer 3 Report

The authors incorporated the suggestions properly. Acceptable!

Author Response

Thanks to dear expert for your meticulous guidance。

Reviewer 4 Report

The authors modified the title of the article and made significant changes in the text that improved the informative value of the article.

In the Introduction chapter, they edited and supplemented the overview of the solved issue, using other relevant sources.

In chapter 4.2, they replaced figure 15 (now 14), which increased the informative value of the performed experiment.

Comments:

1. Please reformulate (explain more clearly) the paragraph (line 428 - 433).

2. In chapter 1 and 2, I did not find a reference to the literature [16], [21] [28,29] and [31].

Author Response

(The authors gave the same response as above.)
